# Uncovering the Host Range–Lifestyle Relationship in the Endophytic and Anthracnose Pathogenic Genus *Colletotrichum*

**DOI:** 10.3390/microorganisms13020428

**Published:** 2025-02-16

**Authors:** Jacy Newfeld, Ren Ujimatsu, Kei Hiruma

**Affiliations:** Department of Life Sciences, Graduate School of Arts and Sciences, The University of Tokyo, Tokyo 153-8902, Japan; newfeld-jacy@g.ecc.u-tokyo.ac.jp (J.N.); ujimatsu-ren@g.ecc.u-tokyo.ac.jp (R.U.)

**Keywords:** *Colletotrichum*, plant–microbe interactions, endophytism, pathogenicity, *Arabidopsis thaliana*, mycoherbicides, fungicides, biofertilizers

## Abstract

*Colletotrichum* includes agriculturally and scientifically important pathogens that infect numerous plants. They can also adopt an endophytic lifestyle, refraining from causing disease and/or even promoting plant growth when inoculated on a non-susceptible host. In this manner, the host range of a *Colletotrichum* fungus can shift, depending on whether it exhibits endophytic or pathogenic lifestyles. Some fungi, such as *Colletotrichum tofieldiae*, can even shift between pathogenicity and endophytism within the same host depending on the environmental conditions. Here, we aim to disentangle the relationship between lifestyle and host range in *Colletotrichum*. Specifically, we aim to demonstrate that lifestyle is dependent on the host colonized in many *Colletotrichum* fungi. We discuss the ways in which pathogenic *Colletotrichum* species may act endophytically on alternative hosts, how comparative genomics has uncovered candidate molecules (namely effectors, CAZymes, and secondary metabolites) underlying fungal lifestyle, and the merits of using endophytic fungi alongside pathogenic fungi in research, which facilitates the use of reverse genetics to uncover molecular determinants of lifestyle. In particular, we reference the *Arabidopsis thaliana*–*Colletotrichum tofieldiae* study system as a model for elucidating the dual roles of plant–fungus interactions, both endophytic and pathogenic, through integrative omics approaches and reverse genetics. This is because *C. tofieldiae* contains closely related pathogens and endophytes, making it an ideal model for identifying candidate determinants of lifestyle. This approach could identify key molecular targets for effective pathogen management in agriculture. Lastly, we propose a model in which pathogenic lifestyle occupies a different host range than the endophytic lifestyle. This will enhance our understanding of pathogenicity and endophytism in a globally significant fungal genus and lay the groundwork for future research examining molecular determinants of lifestyle in plant-associated fungi.

## 1. Introduction

 In nature, plants interact with diverse microbes. These can include pathogens (plant-detrimental), commensals (plant-neutral), and mutualists (plant-beneficial) [1]. Commensal and mutualistic microbes that live inside host tissues without visible disease symptoms for at least a portion of their lives, termed endophytes, can serve diverse functions for the host plant, including protection against pathogens [2,3], nutrient acquisition [4], priming of the immune system [5], and improvement of abiotic stress tolerance [6]. The term endophyte as we use it here is distinct from the term non-pathogen, which refers to any microbe that does not colonize a host pathogenically. For example, a microbe that is unable to infect a host at all may be termed a non-pathogen of that host, but it would not qualify as an endophyte unless it colonizes the host and its re-isolation and colonization within the host have been thoroughly demonstrated. Pathogens cause plants to lose biomass and nutrients and cause severe yield loss annually to important crop species [7,8,9]. At the same time, it has been gradually recognized that some pathogens can act endophytically depending on the host colonized [10,11], or within the same host but depending on certain conditions, such as the stage of the interaction [10] or the environment (e.g., temperature [12]). It has been established that the conditions of the plant–microbe interaction, such as the environment or the host genotype, can affect the outcome of the interaction [13,14]. For example, elevated temperature can suppress effector-triggered immunity (ETI) in plants and cause pathogens with avirulent effectors to become more virulent [15,16]. Other environmental variables have also been demonstrated to be important in shaping the plant–microbe interaction. For example, rainfall has been positively correlated with the proportion of diseased flowers in *Citrus* × *sinensis* anthracnose caused by *Colletotrichum acutatum* [17]. While related to atmospheric humidity, rainfall may have distinct effects on the plant–microbe interaction due to the direct contact between plant tissues and water droplets. As another example, pH has been demonstrated to affect virulence in *Colletotrichum* [18,19]. By modulating transcription factor gene expression, pH altered the fungus’ ability to transition from biotrophy to necrotrophy in the host plant [18]. As a final example, increasing light intensity transitioned foliar endophytic *Diplodia mutila* into a pathogen, potentially due to increased melanization and a faster growth rate associated with more light [20]. Additional variables, such as the atmospheric humidity, soil moisture, soil nutrient content (see Section 4.2), or presence of other microbes, may also be important in the plant–microbe interaction. While the environment has been demonstrated to contribute to plant-associated fungal lifestyles, there is relatively less discussion of the fact that the conditions of the interaction can also determine the lifestyle of a microbe as it colonizes its host. Lifestyle, as we use it here, refers to the fungus’ relationship with its plant host (i.e., pathogen, commensal, mutualist, endophyte). To the observer, this relationship is categorized based on the plant phenotype (e.g., diseased lesions, plant growth effects). As an example, *C. acutatum* was isolated from strawberry, then inoculated onto various crop plants (strawberry, eggplant, tomato, and pepper) [11]. The fungus was able to survive and produce conidia on all plants, but only caused disease on strawberry, demonstrating the role of the host in determining the outcome of the plant–microbe interaction [11]. However, there has not been any unifying understanding of the relationship between host range (i.e., the hosts a microbe can colonize) and microbial lifestyle (e.g., endophyte, pathogen). Additionally, the molecular mechanisms underlying diverse lifestyles ranging from pathogenic to endophytic remain poorly understood. Such questions are particularly important for agriculture, where distinguishing pathogens from endophytes remains an important consideration, as mis-targeting endophytes as pathogens would limit the beneficial effects of endophytes, and mis-targeting pathogens as endophytes would promote microbes that inhibit the growth of or even kill important crop species.

One particularly important plant-associated microbial genus is *Colletotrichum*, which comprises scientifically and agriculturally relevant fungi. Indeed, *Colletotrichum* species were ranked in the top ten most important fungal pathogens in molecular plant pathology, based on their use in science as model organisms and their importance as pathogens in agriculture [21]. At the genus level, *Colletotrichum* fungi can infect a wide range of host plants, including a number of monocotyledonous and dicotyledonous plants, generally causing a disease called “anthracnose” [22]. Anthracnose is characterized by dark necrotic lesions appearing on a variety of plant tissues, ranging from cotyledons on seedlings to fruits [22]. However, not all *Colletotrichum* species cause diseases on all hosts in their host range. For example, *C. tofieldiae* str. 4 colonizes *Arabidopsis thaliana* root tissues without causing any visible disease symptoms, and instead actually promotes plant growth [12]. *Colletotrichum* fungi have been isolated from various hosts such as *Spinacia* species (*C. spinaciae*), *Lilium* species (*C. spaethianum*, *C. lilii*), *Phaseolus* species (*C. lindemuthianum*, *C. nymphaeae*, *C. plurivorum*, *C. sojae*, *C. spaethianum*, *C. phaseolorum*), and *Triticum* species (*C. karsti*, *C. graminicola*, *C. cereale*) [22,23,24,25]. When causing disease, *Colletotrichum* fungi can be devastating pathogens that can affect plants at all developmental stages [22]. Indeed, anthracnose is commonly reported on major crop species such as potatoes, chilis, cereals, and soybeans [22,23]. *Colletotrichum* fungi also have a diversity of lifestyles, including both endophytes, such as *C. tofieldiae* str. 61 [26] and str. 4 [12], and pathogens, such as *C. higginsianum* [27] and *C. incanum* [22], although as discussed later, lifestyle can depend on the host colonized (Section 2; Figure 1A) and the environment (Figure 1B). In this way, *Colletotrichum* makes an effective system for studying host–microbe interactions and microbial lifestyle because closely related endophyte–pathogen pairs may be analyzed in terms of the molecular differences between them. A single fungal species or strain may also be studied on different hosts to identify the factors that differentiate the lifestyle on one host from another. Therefore, this genus makes an excellent model for studying the determinants of lifestyle in plant-associated fungi.

In this review, we aim to disentangle the relationship between lifestyle and host range in endophytic and pathogenic fungi, with a particular focus on *Colletotrichum* species. We discuss how lifestyle is a consequence of the host colonized, with the same fungal species being endophytic on a largely different range of hosts than on which it is pathogenic. We then discuss progress made in elucidating candidate molecular determinants of lifestyle using comparative genomics. We next evaluate the benefits of applying endophytes to phytopathology research through the use of comparative omics and reverse genetics. In particular, we reference the *A. thaliana–C. tofieldiae* study system. This makes a fascinating system to study the determinants of lifestyle and to understand the relationship between lifestyle and host range because endophytic and pathogenic *C. tofieldiae* strains have close phylogenetic relationships. We then note the benefits of applications of *Colletotrichum* research to agriculture, including the benefits of endophytes and the lessons we have learned from pathogens. In particular, in agriculture, it is important to consider host range when combatting potential pathogens. This is because many pathogens adopt an endophytic lifestyle when they colonize alternative hosts, and many endophytes adopt a pathogenic lifestyle when they colonize alternative hosts. In this way, lifestyle is dependent on host colonization, so the elimination of “pathogens” may result in reduced endophytic fungal growth, and the promotion of “endophytes” may result in enhanced pathogenic fungal growth. Therefore, host range must be well understood in order to most effectively combat pathogens and promote endophytes in fields. Lastly, we propose a model for the relationship between host range and lifestyle and discuss future directions for plant–fungus interaction research.

## 2. Lifestyle as a Result of Host Colonization

The lifestyle of plant-associated fungi can be considered in terms of the host colonized. Recent accumulating reports suggest that many fungi traditionally considered pathogens actually behave endophytically on alternative hosts [11,28,29,30]. Likewise, there are reports suggesting that some endophytes can behave pathogenically on alternative hosts. As one example, *C. tofieldiae* str. 4 was originally isolated from diseased *Ornithogalum umbellatum* [31] but behaves endophytically on *A. thaliana* [12]. It behaves endophytically on *A. thaliana* at least partially due to a lack of expression of a putative abscisic acid and botrydial secondary metabolism biosynthesis gene cluster called ABA-BOT, which is activated in a closely related pathogenic strain of *C. tofieldiae* (see Section 4.2.) [12]. As another example, foliar endophytic *Colletotrichum* strains originally isolated from an invasive plant species display virulence on several native plant species in China [32]. These reports highlight that endophytic fungi may behave pathogenically, depending on the host they colonize. For the most part, fungal pathogens occupy a narrow host range, a phenomenon that has been labeled “host species specificity” [33]. From the host plant’s perspective, this is due to “non-host resistance”, a durable resistance system that protects most plants from most pathogens [34]. While out of the scope of this review, it is nonetheless important to note that the host plant plays an active role in the plant–microbe interaction. Conversely, the host range of endophytes tends to be different than the host range of pathogens, frequently encompassing taxonomically distantly related species [35,36]. In this section, we discuss the host ranges of endophytic and pathogenic fungi.

### 2.1. Host Range of Pathogens

Generally, fungal plant pathogens can infect only a limited number of host species [34]. Some *Colletotrichum* pathogens also occupy a limited host range, whereas others actually occupy a broad host range [22,23,37,38,39] (Table 1). Indeed, some species exhibit little-to-no host specificity (e.g., [37,39]), whereas others are highly host-specific (e.g., [38]). For example, *C. lupini* is restricted to *Lupinus* species pathogenically [40]; *C. boninense sensu lato* has a broad host range as a species complex, but within this species complex, some species show restricted host ranges [41]; some isolates of *C. capsici* were shown to be restricted in disease-causing growth to certain legumes, although at the species level, the host range is known to be broader [42]; *C. acutatum sensu lato* as a species complex infects a number of host plants, but certain species within the species complex exhibit host specificity [43]; *C. graminicola* is restricted to *Zea mays* and *Sorghum* [38,44]; *C. orbiculare sensu stricto* is restricted to Cucurbitaceae and *Nicotiana* species [45,46]; *C. higginsianum* infects Brassicaceae plants pathogenically and has also been isolated from diseased *Campanula* and *Rumex* species [27,47,48]; and *C. siamense* pathogenically infects *Capsicum* species and the phylogenetically distant *Pyrus pyrifolia* [49]. At the same time, however, *C. siamense* occupies an entirely different host range as an endophyte, highlighting that one species can act endophytically on some hosts while behaving pathogenically on others [50,51]. It is important to note that *C. siamense* infects multiple hosts both as a pathogen and as an endophyte, underscoring its ability to inhabit a particularly broad host range. While there is a paucity of evidence available about the mechanisms of its dual lifestyles, either pathogenic or endophytic depending on the host colonized, it is known that *C. siamense* produces secondary metabolites that likely function in virulence on rubber trees [52]. However, one volatile organic compound produced by *C. siamense* is a known plant growth-promoting compound, which one could speculate may be related to its endophytic lifestyle on other hosts [52]. Species-level discussion, however, is problematized when one considers the strain-level specificity of lifestyle on the same host. For example, some *C. fructicola* isolates are endophytic on *Coffea* species, whereas others are pathogenic [53]. Likewise, *C. tofieldiae* str. 3 is pathogenic on *A. thaliana* whereas *C. tofieldiae* str. 4 and str. 61 are endophytic [12,26]. Similarly, *C. brasiliense* has been isolated from *Passiflora edulis* as an endophyte and as a pathogen, depending on the isolate [41]. However, strain-level information is not available for all *Colletotrichum* species in the literature, as the taxonomy is frequently only presented down to the species or species complex level. Therefore, we continue our discussion with the consideration of taxonomy down to the strain level where available, and to the species level otherwise.

### 2.2. Host Range of Endophytes

Endophytes, as opposed to pathogens, live within host tissues without causing disease symptoms, and may provide benefits to the host plant, observable on the macro scale as a plant growth promotion phenotype. As opposed to the narrow host range exhibited by many, but notably not all, pathogens, fungal endophytes frequently have a broad host range [35,54,55]. Because the number of studies empirically testing endophytism (i.e., the isolation of a fungus from non-diseased tissue, re-inoculation of the host to validate a lack of disease symptoms, and microscopic visualization to verify that the fungus grows within host tissues) is extremely limited, especially for *Colletotrichum* fungi, here, we consider fungi that have been isolated from surface-sterilized, non-diseased tissues as endophytes. As a rare example of a well-characterized endophyte [26], *C. tofieldiae* str. 61 is able to colonize and promote growth of distantly related tomato and maize plants after being isolated from *A. thaliana* [56,57]. Many species, however, are endophytic or pathogenic depending on the host colonized [11,28,29,30,32,50] (Table 1). As one example, *C. boninense sensu lato* has just one genus, *Coffea*, represented by both endophytic and pathogenic lifestyles [41]. Additionally, *C. chrysophilum* has been reported as a pathogen of banana, cucurbits, and cashew [58,59,60], but other isolations from symptomless plants imply that this fungus is also an endophyte of unwounded cashew, *Theobroma*, and *Genipa* [61,62]. A fourth example can be found in *C. coccodes*, which pathogenically infects potatoes, tomatoes, and chilis, but endophytically colonizes cabbage, white mustard, lettuce, and chrysanthemum [25,63,64]. Acknowledging these findings, it is apparent that lifestyle is dependent on the host colonized for many *Colletotrichum* fungi, with the endophytic lifestyle having a largely distinct host range compared to the pathogenic lifestyle.
microorganisms-13-00428-t001_Table 1Table 1A summarization of reported hosts of *Colletotrichum* fungi as pathogens and as endophytes. Underlined species are shared hosts of the fungus as a pathogen and as an endophyte. Note that not all references in the table are present in the main text.*Colletotrichum* SpeciesHost Range as PathogenHost Range as Endophyte*C. lupini**Lupinus* species [40]*Coffea arabica* [65]*C. boninense s. l.**Protea* species, *Carica papaya*, Capsicum species, *Citrus* species, *Coffea *species, *Cucurbita* species, *Cymbidium* species, *Dracaena marginata, Diospyros australis, Solanum* species, *Passiflora edulis*, *Crinum asiaticum, Euonymus japonica, Persea americana, Phyllanthus acidus, Brassica oleracea* [41]*Maytenus ilicifolia*, Podocarpaceae, Myrtaceae, *Coffea *species, *Mangifera indica*, *Pachira* species *Pleione bulbocodioides*, *Oncidium flexuosum, Quercus salicifolia, Theobroma cacao, Zamia obliqua, Parsonsia capsularis* [41]; *Eperua falcata*, *Goupia glabra, Manilkara bidentata, Mora excelsa, Catostemma fragrans, Carapa guianensis* [66]; Podocarpaceae [67]*C. capsici**Piper betle*, *Vigna unguiculata, Phaseolus vulgaris, Cicer arietinum, Lupinus angustifolius, Pisum sativum, Vigna* species [42]*Arachia hypogaea*, *Cajanus cajan*, *Phaseolus vulgaris, Phaseolus lunatus* [42]*C. acutatum s. l.**Malus domestica*, *Aspalathus linearis, Acer platanoides, Araucaria excelsa, Coffea arabica, Cyclamen* species, *Fragaria* x *ananassa, Lobelia* species, *Chrysanthemum coronarium, Actinidia* species, *Anemone coronaria, Cyclamen* species, *Hevea brasiliensis, Lupinus* species, *Nuphar luteum, Nymphaea alba, Oenothera* species, *Passiflora edulis, Parthenocissus* species, *Penstemon* species, *Persea americana, Prunus cerasus, Pyrus* species, *Solanum* species, *Ugni molinae*, *Vaccinium corymbosum*, *Citrus* species, *Mahonia aquifolium* [43]*Coffea robusta*, *Mangifera indica, Tsuga canadensis, Theobroma cacao, Dendrobium nobile* [43]*C. graminicola**Zea mays* [44]; *Sorghum* species [38]*Zea mays* [68]; *Lolium perenne* [69]*C. orbiculare s. s.*Cucurbitaceae species [45]; *Nicotiana* species [46]
*C. higginsianum*Brassicaceae species [27]; *Campanula* species [48]; *Rumex* species [47]*Centella asiatica* [70]*C. siamense**Capsicum* species, *Pyrus pyrifolia* [49]; *Hevea brasiliensis* [71]*Pennisetum purpureum, Cymbopogon citratus* [50]; *Paullinia cupana* [51]*C. fructicola**Coffea* species [53], *Camellia sinensis*, *Citrus* species, *Malus* species, *Persea americana*, *Prunus persica, Pyrus* species [72]; *Hevea brasiliensis* [71]*Coffea* species [53]; *Hevea* species [73]; *Dendrobium* species [74]*C. tofieldiae**Arabidopsis thaliana* [12], *Ornithogalum umbellatum* [31]; *Capsella rubella* [26]*Arabidopsis thaliana * [12,26,57]; *Zea mays*, *Solanum lycopersicum* [56]; *Cardamine hirsuta* [26]*C. brasiliense**Passiflora edulis* [41]*Passiflora edulis* [41]*C. chrysophilum*Cucurbitaceae [60], *Musa* species [58]; *Anacardium occidentale* [59]*Theobroma* species, *Genipa* species [62]; *Anacardium occidentale* [51]*C. coccodes**Capsicum* species [25]; *Solanum lycopersicum, Solanum tuberosum* [63]*Brassica alba, Latuca sativa, Lepidum sativum, Brassica oleracea, Chrysanthemum* species, *Solanum nigrum* [63]


## 3. Comparative Genomic Approaches for Understanding the Host Range of Pathogenic *Colletotrichum* Fungi

Since the question of how the host range is shaped in host–microbe interaction has attracted many researchers, numerous works attempting to answer this question have been conducted [75,76]. Especially in phytopathogenic fungi and oomycetes, genetic and genomic approaches have expanded our understanding of what kind of genes have evolved and are utilized to invade host plants [77,78,79,80]. As mentioned above (Section 2), *Colletotrichum* fungi have notable diversity in host range and lifestyle [22,81]. Comparative genomic studies have been conducted to uncover the origin and the causes of their diverse host range and lifestyles.

In O’Connell et al.’s work [82], the authors for the first time reported the genomes of two pathogenic *Colletotrichum*, namely *C. higginsianum* and *C. graminicola*, and compared their genomes. *C. higginsianum* has a larger and more diverse repertoire of candidate secreted effector genes (365 genes, 72% of the genes are species-specific) than *C. graminicola* (177 genes, 48% are species-specific). They discussed that the expanded effector repertoire in *C. higginsianum* might result from a broader host range of *C. higginsianum* (see also Section 2.1). Furthermore, genomic analyses showed that the genes encoding pectin-degrading enzymes among carbohydrate-active enzymes (CAZymes) are more than twice as abundant in *C. higginsianum* compared to *C. graminicola*, and the majority of them are expressed in the necrotrophic infection stage. Transcriptome analyses of both fungi during infection revealed that *C. graminicola* upregulates cellulases and especially hemicellulases. This tendency might reflect the difference in cell wall composition between dicots, which have enriched pectin, and monocots, which contain more hemicellulose. Another comparative genomics study following O’Connell et al. indicated that secreted proteases, such as subtilisins, are expanded in *C. orbiculare*, *C. fructicola* (previously identified as *C. gloeosporioides*), and *C. higginsianum*, compared to other hemibiotrophic fungi, such as *Magnaporthe oryzae* and *Fusarium graminearum* [82,83]. Especially in *C. orbiculare* and *C. fructicola*, metalloproteases are also highly expanded, suggesting their role in invasion into the host plant tissue [83]. Interestingly, there are slight differences in the number of genes encoding CAZymes between *C. orbiculare* and *C. fructicola* (327 and 364, respectively), despite their distinct host ranges—*C. orbiculare* exhibits a narrow host range, whereas *C. fructicola* has a broader host range [83]. These reports highlighted that *Colletotrichum* species show similarity in some aspects despite their relatively distant relationship within the genus. However, the expression landscape reported is dynamically different among species, implying that transcriptome plasticity determines the host range and/or fungal lifestyle as a pathogen [82,83]. Supporting this idea, it has been reported in a pathosystem that transcriptional differences upon host defense may shape different host ranges between close relatives of *Sclerotinia* spp. [84].

Insights into the genomic evolution of *Colletotrichum* fungi during host adaptation have also accumulated further in recent years through genomics and/or transcriptomics approaches. For example, Gan et al. [85] sequenced *C. incanum*, which can infect both dicot (*Raphanus sativus* and *A. thaliana*) and monocot (lily) plants, and conducted genus-wide comparative analyses together with five other *Colletotrichum* species, each predominantly associated with either monocot or dicot hosts. This study aimed to find out hints at how *Colletotrichum* pathogens can infect diverse plant species. Interestingly, their analyses for identifying genes with positive selection in those genomes revealed that *C. incanum* genes encoding candidate secreted or nuclear-localized proteins, which are highly conserved among *Colletotrichum* species, are predicted to undergo diversification [85]. This analysis suggested that functional mutation or polymorphism in these gene families contributes to niche adaptation [85]. However, the study also revealed that gene family losses are predicted to occur more frequently than gene family gains in most pathogenic *Colletotrichum* species, including *C. incanum*. These seemingly contradictory findings suggest a complex evolutionary history shaped by genomic adaptation [85]. They also conducted comparative genomics among the *C. gloeosporioides* species complex and found that effector gene clusters were associated with telomeres and repeat-rich regions, suggesting the host adaptation along with the evolution of an accessory genomic region [86].

Baroncelli et al. [87] conducted a genus-wide comparison focusing on the difference between dicot-infecting and monocot-infecting *Colletotrichum* fungi. The study highlighted that monocot-infecting species evolved from dicot-infecting ancestral *Colletotrichum* through independent host jumps accompanied by a decrease in CAZyme repertoires. From comparative transcriptomics, they found that several core genes, including transcription factors, are potentially required commonly to adapt to plant niches, whereas each gene expression profile showed high diversity. Genomic variation is even observed within a single species between *C. higginsianum* strains, particularly in gene-poor regions (often referred to as accessory regions) where candidate effectors and secondary metabolism clusters are located. The variation is suggested to be associated with transposable elements [88,89]. In further research, comparisons among *C. graminicola* isolates detected high polymorphism in both coding and non-coding regions of virulence-related genes [90]. Together, large-scale genomics has provided significant insights into the diversification of the *Colletotrichum* genome, particularly in accessory genomic regions predicted to contain virulence-related genes. This variation may play a crucial role in shaping the host range. These analyses also provided candidate gene families for future investigation into the molecular mechanisms of host adaptation.

To date, how host range and specificity are formed remains a big question; however, Inoue et al. [91] identified effector genes potentially responsible for host specificity utilizing transcriptomics and the generation of multiple candidate gene knockout mutants. *C. orbiculare* infects cucurbits and distantly related *Nicotiana benthamiana.* They compared fungal gene expression profiles during cucumber infection and *N. benthamiana* infection and found four cucurbit-specific effectors that were highly expressed in the early infection stage and required for virulence on cucumber and melon hosts. Notably, the expression of these effector genes was less enhanced in *N. benthamiana* and dispensable for infection. This study highlighted the possibility that *Colletotrichum* fungi use diverse sets of effector genes in response to different host plants, resulting in successful colonization.

Overall, while genomics and transcriptomics have revealed the genomic diversity of *Colletotrichum* fungi, how these variations shape different host ranges remains to be elucidated. Integrating these approaches with genetic analyses would serve as powerful tools for uncovering the molecular determinants of the *Colletotrichum* host range. However, selecting an appropriate *Colletotrichum*–host infection system remains a challenge. The extensive diversity in gene repertoires and transcriptional landscapes across different *Colletotrichum*–host interactions presents numerous candidate genes potentially influencing the host range, complicating systematic investigations.

## 4. Merits of Applying Endophytic Fungal Research to Phytopathology

While phytopathology has traditionally focused on pathogenic organisms, incorporating endophytes into research provides significant advantages. Since both pathogens and endophytes colonize plant tissues, comparative analyses can differentiate factors essential for general host colonization from those specifically required for pathogenesis, thereby identifying key virulence determinants that are absent or unexpressed in endophytes. Notably, examining closely related pathogens and endophytes offers valuable insights into the molecular basis of their contrasting lifestyles. Such reverse genetics approaches are especially useful in *Colletotrichum*, where sexual reproduction is rarely observed, and there are strain-level differences in thallism—i.e., the ability to sexually reproduce within a single individual (homothallism) or whether multiple individuals are required (heterothallism)—so forward genetics approaches are difficult to accomplish [92]. Here, we discuss the merits of applying endophytes to phytopathology, with particular reference to *C. tofieldiae* as a system for studying fungal lifestyle.

### 4.1. Comparative Omics

One way to approach identifying molecular determinants of lifestyle is through comparative omics, as partially outlined in the preceding section with a focus on pathogenic species. Through this approach, endophytes and pathogens are directly compared, and differences between them are identified (e.g., [93]) (see also Section 3). Typically, comparisons are made only between strains of similar lifestyles (e.g., [94,95,96,97,98], but see also [12,99]), although sometimes of varying virulence levels (e.g., [100,101]) or style of infection (e.g., biotrophy vs. necrotrophy) [82] (see also Section 3 for pathogen–pathogen comparisons). While this has nonetheless led to interesting findings, there is a largely unexplored plethora of information that can be uncovered through comparing fungi of different lifestyles. In particular, comparing closely related endophyte–pathogen pairs may overcome the traditional difficulties of reverse genetics where single-gene knockouts do not yield a phenotype different from the wild type. Due to the close phylogenetic relationship between such strains, there will likely be relatively few genes that differentiate them; thus, the candidate gene list will be relatively shorter. In this way, such comparisons may lead to the identification of specific genes that contribute to endophytism or pathogenicity.

For example, Hiruma et al. [12] compared the genomes and transcriptomes of two closely related *C. tofieldiae* strains with remarkably different lifestyles. Str. 3 is a pathogen, whereas str. 4 is an endophyte, on *A. thaliana*. Using reverse genetics, they identified a putative abscisic acid and botrydial biosynthesis cluster (ABA-BOT) uniquely activated in the pathogenic *C. tofieldiae* str. 3 that contributes to pathogenesis. When knockout mutants of genes in ABA-BOT were generated, they grew endophytically and promoted plant growth under low-phosphate conditions, similar to other endophytic *C. tofieldiae* strains, indicating a significant contribution of this gene cluster to pathogenesis.

As another example, Fu et al. [102] compared two strains of *C. gloeosporioides*, one pathogenic and one endophytic. They identified pathogen-specific gene clusters, which were enriched in polyketide synthase genes, potentially suggesting that polyketide synthases are important regulators of lifestyle in *C. gloeosporioides*.

As a third example, Hacquard et al. [103] compared the genomes and transcriptomes of endophytic *C. tofieldiae* and pathogenic *C. incanum*. Although the overall gene quantity was similar across *C. tofieldiae* and *C. incanum*, and the majority of genes are orthologous between the two species, *C. tofieldiae*-specific genes were enriched in secondary metabolism biosynthesis genes, and the number of species-specific effector genes was smaller than those in *C. incanum*. They then found that *C. tofieldiae* has more chitin-binding CAZymes than *C. incanum*, although during the colonization of *A. thaliana*, *C. tofieldiae* does not activate many CAZymes, effectors, or secondary metabolism biosynthesis genes, especially in the early stages of colonization. Broadly, Hacquard et al. found genomic and transcriptomic signatures of lifestyle by comparing the genomes and transcriptomes of closely related *Colletotrichum* species. These studies demonstrate the potential utility of analyzing endophytes alongside pathogens in identifying determinants of lifestyle. However, there are few systems in which pathogens and endophytes are closely related enough to make meaningful comparisons. When endophytes and pathogens are too distantly related, there are numerous genes, transcripts, and proteins that differentiate them, making direct comparisons difficult. For this reason, we argue that the *C. tofieldiae* study system has been used as a successful model and should be broadly exploited for studies on endophytism and pathogenicity.

### 4.2. C. tofieldiae Study System

*C. tofieldiae* is a species of *Colletotrichum* belonging to the Spaethianum species complex, causing anthracnose in *Agapanthus*, *Lupinus*, *Semele*, and *Tofieldia* species [81,104]. It has also been identified as an endophyte on *A. thaliana*, promoting plant growth and transferring phosphorus to its host under low-phosphate conditions [26]. There are, however, strain-specific lifestyles in this species within the same host: *C. tofieldiae* str. 3 causes disease whereas strs. 4 and 61 promote plant growth in *A. thaliana* under low-phosphate conditions [12]. Notably, strs. 3 and 4 are sister strains and share over 98% of their genomes [12]. Our recent results have revealed multiple interesting findings using the *A. thaliana*–*C. tofieldiae* study system. First, Hiruma et al. [12] found that ABA-BOT makes a significant contribution to *C. tofieldiae* str. 3 pathogenicity. It is uniquely activated in str. 3 and not strs. 4 or 61, although it is present in the genomes of all three strains. Even in str. 3, ABA-BOT expression is dependent on the environment and time point after infection [12,105]. In particular, str. 3 promotes plant growth at 26 °C, and concomitantly, there is a decrease in the expression level of ABA-BOT. This lifestyle transition to endophytism of *C. tofieldiae* str. 3 was dependent on *A. thaliana PHR1/PHL1*, as str. 3 inhibited plant growth at 26 °C in *phr1phl1* mutant plants [12]. This means that, in addition to the host range consideration (Section 2), there is also strong environmental and host genotype dependency in the lifestyles of plant-associated fungi (Figure 1B) (reviewed in [14]).

However, the regulatory mechanisms of ABA-BOT remained elusive until Ujimatsu et al. [105]. Ujimatsu et al. uncovered the function of *BOT6*, a transcription factor, in regulating ABA-BOT and other virulence-related genes. Notably, the overexpression of *BOT6* was sufficient to transition *C. tofieldiae* str. 4 from a root endophyte into an anthracnose leaf pathogen. This is likely due to a combination of overexpressing ABA-BOT, which is known to contribute significantly to virulence in *C. tofieldiae* str. 3 [12] and is indispensable for virulence in the overexpression mutant [105], as well as the overexpression of CAZymes, secondary metabolite gene clusters, and predicted effectors. Although these findings have elucidated important mechanisms of virulence in *C. tofieldiae*, there are still numerous unanswered questions, such as how *BOT6* is regulated, what other genetic determinants of lifestyle exist in *C. tofieldiae* and other *Colletotrichum* fungi, and the specific mechanisms by which ABA-BOT promotes fungal virulence. Because of the high similarity between *C. tofieldiae* str. 3 and str. 4 at a genomic level, one could readily compare the genomes of these fungi to identify candidate determinants of lifestyle. These could then be compared to the genomes of other fungi, and orthologous genes could be identified, which may also be determinants of lifestyle in those fungi (e.g., the presence of ABA-BOT in other fungi apart from *C. tofieldiae*). This would allow discoveries made using the *C. tofieldiae* system to be directly applicable to other plant-associated fungi, including both *Colletotrichum* and other fungi. For example, an ABA-BOT cluster similar to the one identified in *C. tofieldiae* is present in phytopathogenic *Diaporthe helianthi*, and separate ABA and BOT clusters are present in *Botrytis cinerea* [12]. This may represent a virulence factor cluster in *D. helianthi* and *B. cinerea*, as it is in *C. tofieldiae* str. 3. This ABA-BOT cluster could be analyzed in *D. helianthi* and *B. cinerea* to determine its expression pattern and contribution to virulence. In a similar way, other virulence factors identified using *C. tofieldiae* could be analyzed in species with orthologous genes. Unlike many other endophyte–pathogen pairs, *C. tofieldiae* strains are extremely closely related to one another, so the effect of redundancy (i.e., multiple genes sharing similar functions) is likely limited, as only a small pool of genes differentiates *C. tofieldiae* str. 3 from str. 4. Based on our previous reports uncovering determinants of lifestyle using *C. tofieldiae*, we therefore suggest that the *C. tofieldiae* study system should be widely used as a model for understanding endophytism and pathogenicity.

## 5. Applications of *Colletotrichum* Research

*Colletotrichum* fungi have been used as models for studying plant pathogens [46,68,82]. However, there are also an increasing number of studies using *Colletotrichum* endophytes for their plant benefits (e.g., [12,26,56,106,107]). Here, we synthesize the literature on both endophytic and pathogenic *Colletotrichum* species to demonstrate that both endophytic and pathogenic fungal research can advance agricultural sciences.

### 5.1. Endophytic Fungi

Endophytic fungal species can be applied to agriculture for their benefits to their host plants. Indeed, *Colletotrichum* endophytes have been demonstrated to promote plant growth and transfer nutrients to their host [12,26,56]. These fungi could be applied to vulnerable crop species to promote yield while reducing the use of traditional chemical fertilizers, which can cause environmental damage [108,109]. Biofertilizers (i.e., fertilizers composed of beneficial microorganisms) can provide comparable benefits to plants as chemical fertilizers and still be profitable to growers [108,109]. Beneficial *Colletotrichum* endophytes could be incorporated into biofertilizers to help promote plant growth and improve crop yield. At the same time, *Colletotrichum* endophytes may also provide protection against pathogens [107]. Therefore, applying biofertilizers composed of *Colletotrichum* endophytes may provide multiple benefits to the host plants. While *Colletotrichum* endophytes are understudied in regard to biofertilizer production, there have been numerous studies examining biofertilizers in other plant-associated fungal endophytes (reviewed in [110]). Future studies examining *Colletotrichum* endophytes for use in biofertilizers will be essential to translate basic endophytism research to agriculture. However, strain selection must be carefully considered in order to avoid mis-applying pathogenic strains on crops, especially considering that many endophytes can behave pathogenically on certain host species. Strain selection can occur following laboratory analyses of the endophytic and pathogenic host ranges of *Colletotrichum* fungi, which will be a critical next step in *Colletotrichum* research. While studies on the host ranges of *Colletotrichum* fungi are gradually accumulating, more studies approaching this issue will be necessary for growers to implement *Colletotrichum* biofertilizers in their fields.

Conditional pathogenicity may, however, be beneficial to field conditions. For example, if a fungus is pathogenic to weed species (see Section 5.2 for further discussion) or otherwise reduces the detrimental effects of weeds [111], but endophytic on crop species, it could provide dual benefits to the host plant by mitigating the negative effects of weeds while promoting beneficial effects for the crop plant. Like pathogens (see Section 5.2 for additional details), some endophytes are capable of producing phytotoxic secondary metabolites (briefly reviewed in [112]). Therefore, metabolites from endophytic fungi could be extracted and applied to weed plants in order to manage weeds in the field.

Additionally, *Colletotrichum* endophytes have been studied for their secondary metabolites. Secondary metabolites can be a source of beneficial substances to humans, such as compounds that inhibit cancer or treat neurodegenerative diseases [113,114,115]. While sometimes phytotoxic, secondary metabolites can also include plant growth-promoting compounds [116]. The role of secondary metabolites in the plant–endophyte relationship is somewhat less studied, particularly in *Colletotrichum* endophytes, but it stands to reason that *Colletotrichum* endophytes may be a source of yet undiscovered secondary metabolites that promote plant growth or improve human health. In support of this hypothesis, 59 putative secondary metabolite biosynthesis clusters were found in the genomes of plant growth-promoting strains *C. tofieldiae* str. 4 and str. 61, including multiple that were upregulated in planta [117].

### 5.2. Lessons from Pathogenic Fungi

Much of *Colletotrichum* research has historically focused on phytopathogenic species, leading to many discoveries about virulence factors produced by *Colletotrichum* pathogens. These virulence factors can be targeted by plant breeders or bioengineers through the addition of their cognate resistance genes [118,119], although such approaches must be undertaken with caution because increasing the resistance gene load can impose fitness costs that reduce plant growth [120]. Alternatively, virulence factors identified in phytopathogenic fungi can be the target of fungicides, especially virulence factors that are conserved across many fungi (e.g., [121]). While effective when applied appropriately, it is highly possible for resistant strains to emerge over time as a result of the selective pressure imposed by fungicides [122,123,124,125]. Therefore, fungicides may not provide an avenue for durable disease resistance, but rather a short-term management strategy, or ideally, one component of an integrated pest management strategy [126,127]. As another potentially more sustainable option, one could develop fungicides that target core processes in fungi. One notable example is fungicides that target melanin biosynthesis, which is ultimately required by fungi to build up turgor pressure in order to penetrate the plant cell wall [128,129,130]. By developing fungicides that target core processes of pathogenesis, it is possible to minimize the risk of fungi evolving fungicide resistance. Therefore, *Colletotrichum* fungi could be studied for their core processes, and fungicides could be developed based on these processes [131]. In these ways, the identification of virulence factors and core processes provides multiple routes for agronomists to deploy resistant crops or otherwise target pathogens in the field.

Additionally, plant pathogenic fungi have been studied for their herbicidal compounds [132]. Because plant pathogens attack their hosts using herbicidal compounds, it is possible to apply pathogens to the field of non-host species to target weeds. *Colletotrichum* fungi have indeed been studied as mycoherbicides. One example is *C. coccodes*, which was applied to soybean and velvetweed and caused a reduction in growth and competitive ability of velvetweed compared to soybean [133]. *C. truncatum* has also been used as a mycoherbicide against hemp sesbania in field conditions, and it caused high mortality of this important weed plant [134]. Because of the restricted host range of pathogens, it is feasible to apply mycoherbicides to the field to limit the growth of weeds without affecting crop species [135]. Some *Colletotrichum* fungi, as discussed above (Section 2.1), have a limited host range too, making them potential candidates for mycoherbicide production.

## 6. Relationship Between Host Range and Lifestyle

Given the above findings, we propose the following model for the relationship between host range and lifestyle: (1) endophytism and pathogenicity are context-dependent, and may change depending on the host and environmental conditions; (2) pathogenicity and endophytism tend to occupy different host ranges compared to each other; (3) within a single fungus with context-dependent lifestyles, the pathogenic lifestyle is likely to be constrained to different hosts compared to the endophytic lifestyle (Figure 1A). In other words, a fungus will likely occupy different host ranges as a pathogen versus as an endophyte. This means that the overall host range of any single fungus is somewhat broader than one might expect if only examining either its endophytic or pathogenic lifestyle. This also means that endophytes isolated from non-diseased tissue may cause disease on other hosts, and pathogens isolated from diseased tissue may not cause disease on other hosts. Notably, neither the host range of pathogens nor of endophytes apparently correlates with host plant phylogeny. For example, some pathogens and endophytes infect both monocots and dicots, which are phylogenetically distant from each other (e.g., *C. incanum*, *C. fructicola*). At the same time, some fungi have environmental dependency on their lifestyles (e.g., *C. tofieldiae* str. 3; Figure 1B). This means that in some environments, they will be pathogenic, but in others, they will behave endophytically. In the case of *C. tofieldiae* str. 3, there is temperature dependency on its lifestyle (Section 4.2) [12]. This means that, in addition to the host range consideration, one must also robustly understand the environments in which a fungus behaves as an endophyte or pathogen. Benefits of applying fungi will therefore be context-dependent; fungi may infect some hosts endophytically and promote plant growth while growing pathogenically and inhibiting plant growth on other hosts. Even within the same host, a fungus may behave endophytically or pathogenically, depending on the environment. The use of fungi in bioproducts thus necessitates an understanding of the host range and environmental contexts of the specific fungus used, and fungi traditionally thought to be pathogens may actually be endophytic, depending on the host plant or environment. In this way, “pathogens” may actually be useful additions to biofertilizers, and “endophytes” may harm their plant hosts, depending on the host on which they are applied or the environmental conditions of the plant–microbe interaction. Therefore, we suggest that the host plant and environmental condition are crucial considerations when developing and deploying antifungal pest management measures or pro-fungal bioproducts.

## 7. Concluding Remarks and Future Directions

In this review, we have demonstrated that there is a previously undiscussed relationship between host range and fungal lifestyle. Generally speaking, a pathogenic lifestyle occupies a different host range than an endophytic lifestyle, even within the same fungal species (Section 2). This is to say, one fungus may be capable of colonizing a broad range of hosts, but only on a select number of those hosts will it cause disease. On the other hosts, the fungus may infect without causing any disease symptoms and may actually promote plant growth. Such lifestyle differences are driven by diverse mechanisms, but candidates frequently include secondary metabolites, CAZymes, and effectors (Section 3). Such determinants of lifestyle may make useful targets for limiting the spread of pathogens in the field, although care must be taken to avoid mis-targeting endophytes. Future studies should focus on understanding the molecular mechanisms underlying lifestyle, especially in useful study systems like *C. tofieldiae*. This could be accomplished through comparative analyses of pathogens and endophytes, then applying reverse genetics approaches to analyze the contributions of particular genes to endophytism or pathogenicity. In particular, using pathogens or endophytes whose lifestyle can change frequently depending on the host or environmental conditions may make particularly useful systems to identify such candidate mechanisms of lifestyle. This is because reverse genetics can often be troublesome due to single-gene knockouts not yielding phenotypes, but in the case of fungi that can frequently change their lifestyle, there may be single genes that act as significant determinants of lifestyle. These genes and their protein products could be analyzed using a variety of methods, such as AlphaFold predictions to identify putative structures or pull-down assays to identify interacting proteins. Additional genetic analyses, such as the construction of phylogenies, would also facilitate understanding the evolutionary history of such genes that contribute to fungal lifestyle. Furthermore, quantifying the host ranges of endophytic and pathogenic fungi to understand whether endophytism or pathogenicity is broader in the host range could yield additional insights into the relationship between host range and fungal lifestyle. This could be accomplished by inoculating fungi on a broad range of hosts and identifying whether disease symptoms are caused. Then, fungi could be re-isolated from (non-)diseased tissues to validate that the fungus was able to colonize host tissues. Further analyses, such as microscopy to visualize the colonization of host cells, would also be interesting as an avenue for future research. Additionally, applied work on developing bioproducts and appropriate fungicides based on the known determinants of virulence and known beneficial endophytes would be helpful in translating the current basic scientific research into agrosystems. Lastly, understanding the ways in which host plants interpret the plant–fungus interaction across fungi with different lifestyles would be helpful to understanding the ways in which host plants discriminate endophytes from pathogens. This would be particularly interesting to explore in closely related endophyte–pathogen pairs or within a single fungus that behaves endophytically or pathogenically depending on the environment, such as the case with *C. tofieldiae* str. 3. Through such an improved understanding of plant–fungus interactions, we can improve crop yield and tackle food security challenges worldwide.

## Figures and Tables

**Figure 1 microorganisms-13-00428-f001:**
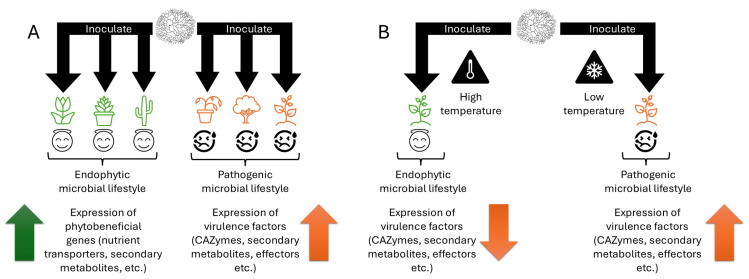
The proposed model of lifestyle in plant-associated fungi. (**A**) Within a single fungus with host-dependent lifestyles, the fungus will likely be endophytic on a different range of hosts than on which it is pathogenic. Thus, if it is inoculated onto a wide range of hosts, it will likely be an endophyte on different hosts than on which it is a pathogen. (**B**) There may be environmental dependency (e.g., temperature) in the fungal lifestyle. A single fungus inoculated under high temperature may therefore behave endophytically even if that same fungus behaves pathogenically under low temperatures (e.g., Section 4.2). We hypothesize that differences in lifestyle may be driven by an increased or reduced expression of virulence factors, such as CAZymes, secondary metabolites, and effectors, or phytobeneficial genes, such as nutrient transporters or phytobeneficial secondary metabolites, although such hypotheses will need to be validated by future research. Note that arrows do not correlate to phylogenetic relationships among host plants.

## Data Availability

No new data were created or analyzed in this study.

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
