# Peer review of "Uncovering the Host Range–Lifestyle Relationship in the Endophytic and Anthracnose Pathogenic Genus Colletotrichum"

_microorganisms, 2025, doi:10.3390/microorganisms13020428_

Round 1

Reviewer 1 Report

Comments and Suggestions for Authors

The manuscript, entitled: "Uncovering the host range-lifestyle relationship in the endophytic and anthracnose pathogenic genus Colletotrichum", organizes our scientific knowledge about the species of a globally significant fungal genus in the framework of a comprehensive study.
The species of the genus Colletotrichum are very diverse. Among them are both plant pathogens and symbionts, and in some cases the same species can show both lifestyles depending on the conditions and the host plant.

Based on all this, in studies, analysis at the strain level is very important in the taxonomy of the Colletotrichum genus. The study analyzes the results of relevant and up-to-date research in sufficient detail.

The manuscript achieves its intended goals, and its execution is almost perfect. It contains a few missing letters and typos. I recommend corrections, e.g. in Table 1, in the case of C. capsici, the name of the host plant Phaseolus lunatus.

After implementing the suggested corrections, I recommend publishing the manuscript as a scientific article.

Reviewer 2 Report

Comments and Suggestions for Authors

The review article titled “Uncovering the host range-lifestyle relationship in the endophytic and anthracnose pathogenic genus Colletotrichum” submitted by Jacy Newfeld , Ren Ujimatsu , Kei Hiruma reports the relationship between lifestyle and host range in Colletotrichum, and discuss the ways in which pathogenic Colletotrichum species may act endophytically on alternative hosts, the molecular mechanisms underlying lifestyle, and the merits of using endophytic fungi alongside pathogenic fungi in research. This is a well-written article and I anticipate that the manuscript should be of great interest to the researchers working on Plant-Microbe Interactions. I consider the manuscript suitable for publication subject to following improvements.

  1. The abstract is well-written and gives a good overview of the subject. It is suggested to revise the aims and purpose of the review, to be more explicit. The description of the Arabidopsis-Colletotrichum system is useful, but a brief discussion of why this system is appropriate for research could enhance the abstract.
  2. The significance of pathogenicity and endophytism in plant-microbe interactions is highlighted in the introduction. A more precise definition of "lifestyle" at the beginning would be helpful, though, given the term is used a lot in the study. Think about rearranging the final paragraph to highlight how important it is to understand host range while combating agricultural pathogens.
  3. The section "Lifestyle as a result of host colonization" presents convincing evidence that Colletotrichum fungi exhibit distinct behaviors depending on the host species. However, the consideration of environmental elements that influence lifestyle might be expanded. The example of temperature influencing ETI is persuasive, but other variables such as humidity or soil conditions may also be relevant.
  4. The section "Host range of pathogens and endophytes" has an extensive and useful explanation of host range. It is important to point out Colletotrichum siamense's ability to inhabit several host ranges as both a pathogen and an endophyte. It would be good to learn more about the molecular mechanisms that are causing this transition.
  5. The comparative genomics section excels, notably in discussing effector genes and CAZymes. Evaluate whether these genetic variations lead to visible phenotypic differences in host colonization. The transition between explanations of various research might be improved by using summary words that tie results.
  6. It is clearly stated how endophytes may be used in agriculture. The fact that not all endophytes consistently promote plant growth, however, may be a drawback. It's intriguing that C. tofieldiae might be used as a model system. This argument would be improved by including a sentence addressing the viability of expanding results from this model to else Colletotrichum species.
  7. The discussion of mycoherbicides and biofertilizers in the "Applications of Colletotrichum research" section is interesting. This section would be improved, though, by more specific examples of effective usage in agricultural contexts. Policymakers and farmers may be quite concerned about the possibility of misapplying pathogenic strains in the field, so this issue needs to be covered in more detail.
  8. Section "Host range and lifestyle relationship" Although the suggested model is intriguing, the text could do an improved job of explaining Figure 1. It would improve understanding if the figure's representation of various host ranges were made clear. If phylogenetic evidence is available, it should be used to further support the hypothesis that endophytic and pathogenic lifestyles may occupy distinct host ranges.
  9. The conclusion makes an excellent task of presenting the main ideas, although it may be improved by highlighting some research gaps. What are the next steps, for instance, to clarify the chemical and biological basis of changes in lifestyle? It would be good to include a brief explanation of possible experimental strategies for evaluating the suggested model.
  10. The overall manuscript is well-written, but some sections contain long, complex sentences. I suggest breaking these sentences into shorter sentences for clarity.
  11. Add latest references from the available literature

Reviewer 3 Report

Comments and Suggestions for Authors

Dear Authors,

I have reviewed your manuscript "Uncovering the host range-lifestyle relationship in the endophytic and anthracnose pathogenic genus Colletotrichum", submitted for publication in Microorganisms.

I find that your manuscript is very well conceptualized, well written, scientifically sound, interesting to read, and overall represents a very useful gift to the readership of Microorganisms. I have truly enjoyed reading your paper and I have gained a lot of new insights on this interesting fungal genus and the logic behind the mechanisms of its pathogenesis. Minor imperfections of your article include, most importantly, several redundant parts of text whereby previously summarized points are again repeated and summarized. At several points, some statements need clarification. For the revision of your paper, please follow my comments as given below:

·         Abstract, penultimate line: please replace "lays" with "lay". Because it is part of future tense ("this will... lay the groundwork...")

·         Introduction:

o    Although the pathogenicity of Colletotrichum is well explained in the Introduction section, specific mentioning of anthracnose as the disease caused by Colletotrichum is missing. A very brief mentioning of the disease itself and brief summarization of its visual characteristics as well as agronomic importance (which crops are most affected and in which ways) would be desirable, especially since the word "anthracnose" is featured in the article title itself.

o    Last paragraph of Introduction, around mid-paragraph: "... we reference the C. toefildiae study system, which makes a fascinating system by which to study..." - please rephrase this sentence to make it more elegant.

·         Lifestyle as a result of host colonization: This part of text contains a repetition of certain points that were already made in the Introduction section, but are not significantly further elaborated here in Section 2. I would suggest that the points concerning the cited references 28, 12, and 29 should be briefly (not too much) elaborated through summarizing some of the most relevant insights from these papers. This is intended mostly so as to prevent the readers from feeling like they are reading an almost identical repetition of the text that they had already read in the Introduction.

·         Host range of pathogens:

o    In this section, unfortunately a disorder in reference numbering has happened. The references 85 and 124 are cited for the first time between the references 36 and 37. I'm afraid that a thorough renumbering of all the references in your paper will be necessary starting from this point. Please make sure that the match between the references cited in the text and those given in the references list, does not become lost when you are renumbering them. I suggest that you do this as a final step before resubmitting your manuscript, when you are sure that no further alteration in the order of reference citing will occur.

o    Another remark concerning this section (as well as the following section) is please double-check that all the examples of pathogenicity or endophytism of Colletotrichum in the diverse range of its hosts, that are mentioned in the manuscript text are featured in Table 1, and vice-versa.

·         Host range of endophytes:

o    line 3: please replace "by" with "as" ("as a plant growth promotion phenotype")

o    in this section, again, the reader gets the impression that a lot of the points made here had already been made in the previous sections of the manuscript. Please take care of these redundancies to avoid making an impression of reading a redundant text.

o    fourth-last line (the line where references [57-59] are being cited): please replace "infects" with "colonizes". Because here you are speaking in terms of endophytism, not infection.

·         Page 6:

o    "and the majority of which are expressed in the necrotrophic infection stage" - please either delete "of which" or replace it with "of them" ("and the majority are expressed" or "and the majority of them are expressed")

o    "during C. higginsiana infection to Arabidopsis thaliana" - please replace with "during C. higginsiana infection of Arabidopsis thaliana"

·         Page 7:

o    line 2: "Interestingly, there are slight differences in gene number..." - Please be more specific about this. Which of the two species had more CAZyme genes? And how is that discussed?

o    Next paragraph, around mid-paragraph: "Conversely, in pathogenic Colletotrichum, ..." - This sentence is somewhat unclear. "Conversely", as opposed to what? The word "conversely" is used when you are announcing something that is kind of opposed to what you have explained previously. But here you are talking about pathogenic Colletotrichum, whereas the previous part of the story concerned C. incanum, which is also a broad-host-range pathogen. Did you maybe mean to say: "Conversely, in most other pathogenic Colletotrichum species, ..."?

o    The sentence starting with "Baroncelli et al. [94]" should be made the beginning of a new paragraph to more clearly suggest the logical structure of the text to the reader.

o    "in intraspecies" - please replace with "within a single species"

o    end of page: "Overall, genomics and transcriptomics can be strong tools..." - This sentence should be separated as a separate paragraph, that would be the closing paragraph of this section. Also, this sentence is very long and it is quite hard to follow for the reader. I suggest to rephrase it and chop it into 2 or 3 shorter sentences.

·         Merits of applying:

o    "(i.e. the ability to sexually reproduce... ... are required (heterothallism))" - here you have a double system of parentheses (parentheses within parentheses). I suggest to use a combined system of parentheses and dashes: "... differences in thallism - i.e., the ability to sexually reproduce within a single individual (homothallism) or whether multiple individuals are required (heterothallism) - so forward genetics..."

o    last line before section 4.1: "as a potential system for studying fungal lifestyle" - Why "potential"? C. tofieldiae has already been used as a successful model system for stydying fungal lifestyle. I would omit the word "potential" because it downplays the actual success of this model system.

·         Comparative omics:

o    The section 4.1. is organized into a single paragraph, for which it is quite difficult to read even though the text is actually very interesting. I suggest to divide it into multiple paragraphs. Each of the narrated examples should be put as a separate paragraph. I suggest three additional paragraphs, one starting with "For example, Hiruma et al. [12]...", then one starting with "As another example, Fu et al. [110]...", and the final paragraph starting with "As a third example, Hacquard et al. [111]..."

o    "... was smaller than those of C. incanum" - I would suggest to replace with "was smaller than those IN C. incanum"

o    "They then found that C. tofieldiae has more chitin-binding CAZymes..." - this sentence talks about "infection" with C. tofieldiae. Is it "infection", or endophytic colonization? Please be clear. Also please pay attention not to use the word "infection" for beneficial endophytic colonization, throughout the manuscript.

o    Last sentence in section 4.1: again, "potential model". I don't think that you can "propose C. tofieldiae as a potential model", since it has been clearly already successfully used as a model. On the other hand, as the authors of this review paper, you can say "we argue" or "we emphasize that the C. tofieldiae study system is a successful model, that should be broadly exploited for endophyte-pathogen comparisons", or something like that.

·         C. tofieldiae study system: "Notably, overexpression of BOT6 was sufficient to transition C. tofieldiae str.4 from an endophyte into a pathogen" - Are there available insights in what kind of effects the overexpression of this gene makes on a molecular-genetic level, i.e., what are the downstream effector genes through which it acts? If there are any more detailed insights into the regulatory effects of BOT6, they should be briefly summarized here.

·         Endophytic fungi, last sentence: "59 putative... clusters were found in plant growth promoting strains..." - were found in the strains, or in their genomes? Please be clear in your statements.

·         Figure 1:

o    The Figure is very appealing and witty, however I find that it should be rendered a bit more informative. In its current form, the only information conveyed by this Figure is that the resolving of the endophyte/pathogen dilemma is dependent on host identity, however, it doesn't offer any other factor that may contribute to its resolving. I suggest that the Figure should be more worked out to make it more detailed and informative, especially since it is the only figure offered in the paper.

o    Figure caption should also be revised. "Proposed idea" is not good wording. Please replace with "proposed model" or some other more suitable wording.

o    The next paragraph ("Within a single fungus...") is formatted like part of the regular manuscript text, but reading it, I have an impression that it actually belongs to the Figure caption. Please double-check and revise as appropriate.

·         References: Please re-number the references so that they match the order of appearance in the manuscript text, and make sure that the renumbered references match their respective citations in the manuscript text.
